# Developing an Evidence-Based Nursing Culture in Nursing Homes: An Action Research Study

**DOI:** 10.3390/ijerph19031733

**Published:** 2022-02-02

**Authors:** Marleen H. Lovink, Frank Verbeek, Anke Persoon, Getty Huisman-de Waal, Marleen Smits, Miranda G. H. Laurant, Anneke J. van Vught

**Affiliations:** 1School of Health Studies, HAN University of Applied Sciences, 6525 EN Nijmegen, The Netherlands; frank.verbeek@han.nl (F.V.); marleen.smits@radboudumc.nl (M.S.); miranda.laurant@han.nl (M.G.H.L.); a.vanvught@han.nl (A.J.v.V.); 2Department of Primary and Community Care, Radboud University Medical Center, Radboud Institute for Health Sciences, 6500 HB Nijmegen, The Netherlands; anke.persoon@radboudumc.nl; 3Scientific Center for Quality of Healthcare (IQ healthcare), Radboud University Medical Center, Radboud Institute for Health Sciences, 6500 HB Nijmegen, The Netherlands; getty.huisman-dewaal@radboudumc.nl

**Keywords:** action research, evidence-based nursing, nursing home, nursing team, organizational change

## Abstract

**Background:** Nursing homes face challenges caused by increasing numbers of older adults with multimorbidity and the demand for quality of care. Developing an evidence-based nursing (EBN) culture is a promising strategy to face these challenges. Therefore, the aim of this study was to develop an EBN culture in nursing homes and gain insight into the influencing factors. **Methods:** An action research study was conducted with 12 nursing teams in 4 Dutch nursing homes, using the Practice Development approach to develop an EBN culture. The teams (mostly certified nurse assistants) were coached by internal facilitators (bachelor’s or master’s degree nurses) and external facilitators (nursing teachers). Data were gathered at baseline and after 15 months using questionnaires and individual and focus group interviews. **Results:** With varying degrees, most nursing teams implemented elements (related to values, attitudes, and behaviors) of an EBN culture with appropriate leadership, advocacy, and training. The team members became open to new insights and asked critical questions. During the project, participants learned how EBN could be incorporated into daily practice, for example, by keeping it small, discussing information from professional journals, and using creative methods such as quizzes. Influencing factors of an EBN culture were: (a) support of managers, (b) inspiring facilitators close to the team, and (c) stable teams with driving forces and student nurses. **Conclusions:** Integrating EBN into daily practice in creative and motivating ways contributes to the development of an EBN culture in nursing homes. To facilitate this, managers should support teams in the process and content of EBN, and internal facilitators should collaborate with driving forces on the teams.

## 1. Introduction

The quality of nursing home care in developed countries is a concern for several reasons [1]. First, the number of older adults with (chronic) diseases and multimorbidity is increasing rapidly, creating a high demand for nursing home care [2]. Second, nursing staff in nursing homes are more likely to have a lower degree of education than nursing staff in hospitals [3]. In the Netherlands, 62% of nursing staff in nursing homes are certified nurse assistants (CNAs) [4]. They follow a two- to three-year training program resulting in European Qualification Framework (EQF) level 3 [5].

Dutch CNAs are not taught to apply evidence-based practice (EBP). Prior research found that a lack of EBP training results in ineffective interventions, and effective interventions are not applied or are rushed [6,7,8]. This negatively affects the quality of care provided and may even harm nursing home residents.

Working according to the principles of EBP is a promising way to enhance the quality of basic nursing care in nursing homes [9]. When referring to nursing practice, the term evidence-based nursing (EBN) is used. EBN is in line with the general definition of EBP: the integration of best research evidence with clinical expertise and resident values when making decisions in nursing practice [10]. EBN should be part of daily practice in nursing homes, and it is important to create an EBN culture in nursing teams [11]. In an EBN culture, reflective professionals work together, support each other, provide each other with the best available evidence, and discuss the best care possible all day, every day.

An organization can support EBN by providing resources and creating a learning culture [12]. In addition, a prerequisite for creating an EBN culture is the presence of transformational (visionary) leaders and EBN facilitators. These roles may be filled by directors, managers, nurses with a master’s or bachelor’s degree, or CNAs [12,13,14]. EBN is further influenced by the quality and accessibility of research findings [12]. While the development of nursing science has accelerated over the past decade, the published research may not always be able to guide nurses in daily practice because it is descriptive, poorly described, or conducted in settings other than nursing homes [15].

In short, developing an EBN culture in nursing homes is quite a challenge. It is known that it requires the implementation of EBN and a transformation of the nursing home organization culture [16,17,18]. However, to our knowledge, there are no models for developing an EBN culture in nursing homes in the Netherlands or in other developed countries, although doing so may improve the quality of care [19]. Therefore, the aim of this study was to develop an EBN culture in nursing homes and to gain insight into the factors that influence an EBN culture in nursing homes.

## 2. Methods

### 2.1. Design

In our EVIDENCE study (Extending Valid Infrastructures to Deploy Evidence in Nursing homes and create a Culture of Evidence), we used an action research design. Action research describes, interprets, and explains social situations while implementing a change intervention aimed at improvement and involvement. The evaluation and change intervention are conducted in partnership with the participants [20]. The study took two years (from May 2017 to May 2019), and the intervention period lasted 15 months.

### 2.2. Setting

Four Dutch nursing homes participated, each with two to four nursing teams. These nursing homes and teams were selected purposively based on the varied skill mix among team staff in order to maximize the transferability of the results to other nursing homes. The participating nursing homes were members of an academic nursing home network aimed at improving the quality of care in nursing homes.

### 2.3. Participants

Twelve nursing teams were the primary study population. The nursing teams included staff educated to all EQF levels (1–7) [5]. Other participants were medical care providers (elderly care physicians, physician assistants, and nurse practitioners), managers, and directors.

### 2.4. Practice Development

In our study, the change intervention was the Practice Development approach (PD). PD is an intervention method based on nine principles: (1) person-centered and different types of evidence, (2) micro-level supported by meso- and macro-levels, (3) learning on the spot, (4) developing and applying evidence, (5) creativity combined with cognition, (6) involvement of stakeholders, (7) tailored methods, (8) facilitating, and (9) involvement of stakeholders in evaluation (Appendix A) [21,22]. Below, we describe how these principles were applied in this study.

The developmental process was facilitated by internal and external facilitators. The internal facilitators were nurses with EQF levels 6 or 7 (e.g., baccalaureate-educated registered nurses, nurse scientists, nurse practitioners) who were employed by the nursing home organization. Each organization appointed one or more internal facilitators to encourage an EBN culture in their nursing teams. Two external facilitators (GB and AR) supported the internal facilitators in the change process by providing relevant information and offering suggestions for practical tools, instruments, or methods. The external facilitators were nursing lecturers at HAN University of Applied Sciences with a background in nursing science and experience in change management. They worked in close collaboration with the action researcher, the academic nursing home network, and the educational nursing programs to optimize the process. Each external facilitator supported the internal facilitators in two nursing homes. In each organization, the internal facilitator(s) had eight hours and the external facilitator had two hours a week for facilitation. Furthermore, the action researcher (M.H.L.) supported the teams by collaborating with the internal and external facilitators, providing the results of the baseline and post-intervention measurements, and searching for relevant evidence-based information for the nursing teams.

Each nursing team chose one or more tailored fundamental nursing care topics to improve by EBN. Examples of topics include: residents’ privacy, getting to know residents on a personal level, oral care, changing urinary catheter bags, changing suprapubic catheters, nurses’ worries or concerns, problem behavior, doctors’ rounds, and do-not-dos (for an example, see Appendix A). Each team worked on two to five topics during the 15-month intervention period. The teams involved the residents and/or their relatives as much as possible (e.g., in identifying the topic and in the evaluation). While working on each fundamental nursing care topic, the nursing teams and the facilitators also worked on meeting the preconditions for an EBN culture, such as knowledge of EBN within the nursing team, involvement of management, and the proper team climate [12,13].

Each organization coordinated the project in the same way. They started the process with a kick-off meeting for the participating teams. The action researcher presented the results of the baseline measurements, and the following points were discussed: the meaning of EBN, how to integrate EBN into daily practice, and fundamental nursing care topics for improvement. Subsequently, each team organized tailored activities such as journal clubs, quizzes, clinical teaching, discussions about resident satisfaction, or a workshop about searching for evidence on the internet. At the end of the project, each organization held a final evaluation meeting with the participating teams, policymakers, managers, and/or directors. All meetings were organized by the internal facilitator, external facilitator, and/or action researcher.

### 2.5. Measurements

We performed a mixed methods data collection with qualitative measures (individual interviews and focus group interviews) and quantitative measures (validated paper-administered questionnaires) at baseline and after 15 months, post-intervention (Table 1).

One experienced qualitative researcher (action researcher MLo) conducted all interviews. Most were face-to-face, but some interviews occurred via telephone.

### 2.6. Analysis

The individual interviews and focus group interviews were audio-recorded. The recordings of the individual interviews were used to summarize each interview. In addition, the post-measurement interviews of the internal facilitators were transcribed verbatim and analyzed through content analysis by open coding and creating categories in Atlas.ti [30]. These interviews were analyzed in-depth, as they contained the most information, namely, information about the facilitation process and changes in EBN culture. Each individual interview was summarized or coded by a bachelor’s nursing student researcher (DS, ES, SJ, SU, SV, and VH) together with the main researcher (MLo). Focus group interviews were transcribed verbatim, and MLo analyzed them in Atlas.ti through open coding and creating categories.

The questionnaires were first analyzed using descriptive statistics in IBM SPSS v24. Inferential statistics were used to compare the results at pre- and post-measurement for the EBPQ-ve and EBPAS-ve because the sample size was greater than 25 for both measurements. We considered 25 to be the minimum sample size for inferential statistics. In the comparative analyses, a multilevel regression analysis in SAS 9.4 was used, with organization as a random effect and controlling for two covariates: number of working hours a week and time since graduation. We controlled for these covariates because these background variables could influence the attitudes and behaviors of individuals regarding EBN.

The results of the quantitative analysis were then linked to main categories that emerged from the qualitative analysis, and those results were compared and integrated. The main categories were described by MLo and first discussed with a second researcher (AvV) and then with the research team (all authors) [30].

### 2.7. Rigor

The rigor of the results was promoted in different manners during this action research study [20]. First, participants with different backgrounds (including nurses, team managers, and medical care providers) were included in the interviews, which contributed to different perspectives on how to develop an EBN culture. Second, both qualitative and quantitative data were collected by different data collection methods. Data and method triangulation can increase the trustworthiness of the results if different methods yield the same results. Thirdly, the key results were fed back to the participant at the start in order to inform their actions. At last, while the project was running, we discussed and validated the preliminary results within the project group, in which participants from the organizations also participated.

## 3. Results

Table 2 describes the EQF levels of the members of the 12 participating teams. Since changes in teams occurred, numbers differed between pre- and post-intervention.

Thirteen internal facilitators participated in the project: ten with EQF level 6 (three received their diploma during the project) and three with EQF level 7. Three internal facilitators quit that role because of a change of job before the end of the project, but they were replaced by three others. One internal facilitator was not a nurse but a physiotherapist.

In total, we conducted 38 individual interviews and 1 double interview with two internal facilitators from the same organization about EBN culture and influencing factors (Table 1). In one organization, no team manager was present, and we could not conduct a post-intervention interview with the contact person for the academic nursing home network due to personnel changes. In another organization, we interviewed a team coach (employed by the organization before this project with a focus on team processes) in addition to the five professionals described in Table 1. Forty-four nursing team members participated in four focus group interviews at baseline. Post-intervention, the four focus group interviews involved 27 nursing team members.

The EBPQ-ve and EBPAS-ve were filled in by 131 nurses (95% women, mean age 42) at EQF levels 1–4 pre-intervention and by 119 (93% women, mean age 37) post-intervention. The EBPQ, EBPAS, and Barriers Scale were filled in by 16 nurses (94% women, mean age 41) at EQF levels 6 and 7 pre-intervention and by 21 (94% women, mean age 38) post-intervention. The LPI was filled in by 12 internal facilitators pre-intervention and by 11 post-intervention.

### 3.1. Categories

Three main categories were identified in the qualitative and quantitative data: (1) EBN culture, (2) EBN integration into daily practice, and (3) factors influencing an EBN culture.

#### 3.1.1. EBN Culture

The interviews revealed that team members did not (consciously) apply all of the principles of EBN when they faced clinical uncertainties, pre- or post-intervention. In particular, they did not always use the best research evidence. However, respondents stated that the nursing team members used protocols and guidelines more often post-intervention.

*“We use more available information since the project started because we need to search for information about the nursing topics we have chosen to improve in the unit”.* (Certified nurse assistant, organization 1)

*“The culture has changed positively. I have the impression that they [nursing team members] figure things out more often and do not accept something as true that easily anymore”.* (Manager, organization 2)

Respondents also stated that a critical attitude and enthusiasm were prerequisites for EBN, but it appeared to take a long time for some team members to develop such an attitude. At baseline, team members’ knowledge and skills related to EBN were reported to be relatively low because most of them had never been taught about EBN. Team members who displayed an open attitude toward EBN at baseline appeared to gain the most EBN knowledge and skills during the project. However, a low level of EBN knowledge and skills among team members was still reported post-intervention.

On the EBPQ(-ve) and EBPAS(-ve), we found few differences between pre- and post-intervention responses. We did note a minor deterioration in EQF level 1–4 nursing team members’ attitudes about EBN and a small increase in the frequency with which EQF level 6 and 7 nursing team members asked residents about their preferences (Table 3).

#### 3.1.2. EBN Integration into Daily Practice

During the project, respondents learned how EBN could be incorporated into daily practice. During the interviews, respondents reported four approaches that motivated nursing team members.

First, the resident and their preferences should be the starting point. For example, the topics for improvement should be chosen after consultation with the resident(s). The nursing team members became enthusiastic when they noticed that their actions contributed directly to residents’ well-being.

Second, it was important for team members to “keep it small”, experience success quickly, and become aware that they sometimes use EBN already, even if unconsciously. Especially the first experience with EBN should be simple and successful.

*“Yes, I think it [EBN] depends on the enthusiasm of the people and the experience that it is effective. […] So, as a facilitator, I can facilitate this process. You address a small topic and make it a success. […] And make this visible, yes. I have learned that maybe we started too big, too abstract, we maybe could have started smaller”.* (Internal facilitator, organization 4)

Third, team members should discuss EBN by talking about the best care possible, integrating EBN into meetings (e.g., care plan meetings), and sharing information that they find in professional journals and on the internet.

Finally, stimulating, innovative, and creative methods of EBN should be used. In one team, for example, the internal facilitators formulated statements about actions in daily care, and the nursing team members had to research various sources to discover whether these statements were true. The team member with the right answer and the best arguments was the winner.

#### 3.1.3. Factors Influencing an EBN Culture

This category consists of the following subcategories: (a) support of managers and research networks, (b) inspiring facilitators close to the team, and (c) stable teams with driving forces and student nurses.

##### 3.1.3.1. Support of Managers and Research Networks

Respondents stated in the interviews that regular meetings between the internal facilitators and their manager(s)/director(s) were very important to create a foundation for EBN. These meetings were facilitated and, in some cases, attended by the external facilitator. The following topics were discussed: the meaning of EBN in daily practice, required and available time and resources, and a vision of EBN in relation to the vision of the organization. An internal facilitator at organization 1 noted: *“Well, I think at first the management has to know how they have to support this whole theme. At the moment, we don’t feel like the manager knows what is going on. […] Of course we catch up with her, but I do not think she has any idea what EBN is, how you should implement it, what is happening in our unit, and what nursing care topics we are working on”*.

The interviews revealed that the nursing teams felt that (research) evidence was not always available or easily accessible. They found it helpful if the organization’s intranet was well-organized and guidelines and protocols were easy to find. In particular, the internal facilitators and contact persons for the research network within the organizations stated that research networks should invest in research evidence that is accessible and transferable to daily practice. As the contact person for the research network at organization 2 stated: *“The knowledge that is provided by the university library does not align with the knowledge of nursing team members”*.

On the Barriers Scale, nursing team members reported the most barriers related to communication and organization (Table 4). Most of them decreased post-intervention. Pre-intervention, the most reported barriers were “there is insufficient time on the job to implement new ideas” (94% moderate or great barrier), followed by “research reports/articles are not readily available” (87%), and “the facilities are inadequate for implementation” (87%). Post-intervention, the top barriers had changed: “the facilities are inadequate for implementation” (74%), “other staff are not supportive of implementation” (65%), “there is insufficient time on the job to implement new ideas” (62%), and “the relevant literature is not compiled in one place” (62%).

##### 3.1.3.2. Inspiring Facilitators Close to the Team

The interviews showed variation in the way that internal facilitators supported the nursing teams. Some only coached the nursing team members (sometimes from a distance), while others were inspiring role models who provided education and collaborated in EBN. For nursing team members, it was essential that the internal facilitator was close to them, literally and figuratively. This meant that the internal facilitator worked in the unit, was part of the nursing team, and was available for questions.

A nursing team member at organization 3 missed the support of the internal facilitator and stated: *“She has never been present on the unit”*. In contrast, a certified nurse assistant at organization 2 stated that the internal facilitator was a role model: “She is baccalaureate-educated and if I have questions, if I ask them, I get answers immediately. I like that”.

The scores of the internal facilitators on the LPI self-assessment were slightly higher post-intervention than pre-intervention (Table 5).

##### 3.1.3.3. Stable Teams with driving Forces and Student Nurses

Respondents stated that the nursing teams in which the EBN culture grew the most had a positive and safe team climate, enough and competent members, and low staff turnover. In those teams, each team member’s roles, tasks, and responsibilities in both daily practice and in this project were clearly defined and communicated. However, it appeared to be difficult to create such a team: (1) because it was difficult to attract and retain nursing team members from all levels and (2) because of the high perceived workload.

The interviews further revealed that teams that were most successful in creating an EBN culture had one or more people who were driving force(s) on their team. These driving forces were very committed to the residents, had a special interest in EBN, and had the ability to enthuse and inspire their team members. Being a driving force was not dependent on education level. An internal facilitator at organization 3 noted: “I think that in that unit [which was less successful at creating a culture of EBN] we missed a driving force. Someone who really goes for it and has a position on the team. Someone who can get the others on board. I think that someone was lacking and is still lacking”.

Finally, respondents stated that the presence of nursing students from all EQF levels on the unit positively influenced the EBN culture. The students asked critical questions during daily care, answered questions by sharing up-to-date knowledge, and performed EBN school assignments in the teams.

## 4. Discussion

This study showed that it is possible to coach nursing teams in nursing homes to work according to the principles of EBN and to create an EBN culture by applying PD. However, not all participating nursing teams were equally successful. The level of success in creating an EBN culture depended on: the support from the organization, the presence of higher educated nurses as inspiring facilitators, and research networks; the stability of the team; and the presence of driving forces and student nurses on the team. Making EBN a part of daily practice in creative and motivating ways contributed to the awareness among nursing team members that EBN is part of their job and not a separate task.

In accordance with another PD study [31] in nursing homes that aimed to improve evidence-based care, our study underlined how important the nine principles of PD are to successfully developing an EBN culture [21]. Going deeper into principle 2 of PD, which deals with micro-level supported by meso- and macro-levels, research by Kaplan and colleagues (2014) in a magnetic hospital showed that managers can be especially important because they can model and promote the use of EBN [32]. For example, they can ask for evidence to support change.

At the start of our study, we assumed that managers and directors had the necessary skills and knowledge to facilitate an EBN culture. However, baseline measurements and first experiences showed this was not the case. Most of the managers had heard about EBN, but they were not exactly aware of its definition, and they lacked the skills to apply EBN and facilitate team members in performing it. Regular discussions between the internal facilitator, external facilitator, and the manager(s)/director(s) helped to jointly create a foundation for an EBN culture.

In addition to the nine principles, the study revealed the importance of having one or more driving forces on the teams to help develop and sustain an EBN culture. The driving force on a team is comparable to the champion role, as described by Woo and colleagues (2017) in their systematic review: a leader who fosters and reinforces changes for improvement [14]. This champion role exists alongside the role of the internal facilitator, and their collaboration strengthens the internal facilitator’s position and actions.

The driving forces in our study took on their role quite naturally. However, no one took that role on some teams, and those teams were less successful in creating an EBN culture. This suggests that selecting or training nursing team members for this role might help in creating an EBN culture. In line with findings from other studies, we found that it was not only higher educated nurses who took this role; in particular, nurses with EQF levels 2–4 were successful driving forces on their teams [14,33,34]. The following characteristics can be used to select or train driving forces: being very committed to the residents, having a special interest in EBN, and having the ability to enthuse and inspire team members [14].

A positive methodological aspect of this study is the data and method triangulation.

We collected robust data by using different methods, at different sites, and from different people. This contributed to a comprehensive picture of creating an EBN culture in nursing homes [35]. The use of different methods also revealed that the results of the questionnaires (EBPQ-ve and EBPAS-ve) showed a more positive image of EBN use, attitude, knowledge, and skills than the findings from the interviews, especially during the baseline measurement. Remarkably, although we saw an improvement during the project, their scores remained the same, probably because they became conscious of their incompetence. We even saw a significantly lower score on the “attitude” subscale post-intervention compared to baseline. There is no unambiguous explanation for these results, but the participants may have given socially desirable answers at baseline [36] or may have been unconsciously incompetent and therefore given relatively higher scores [37].

The primary aim of this study was to develop an EBN culture. Therefore, the measurements were focused on the EBN culture and not the results of the culture change. However, it would be interesting to also include the effects of the change in EBN culture in follow-up research, for example, on nurse-sensitive outcomes. This does require a different approach in order to be able to collect sufficient data at the patient level.

In this action research study, the role of researcher and facilitator were separated, in contrast to most action research studies in which these roles are performed by the same person [20]. In our study, the action researcher performed the measurements and presented the results together with the external facilitators to the teams. The external facilitators supported the internal facilitators, driving forces, and teams in the change process. In this manner, the action researcher was not a “co-constructor” of data, which contributed to the trustworthiness of the findings [38].

## 5. Conclusions

It is challenging to create an EBN culture in nursing homes, but by applying the principles of Practice Development, most nursing teams succeeded to a greater or lesser extent. The nursing teams that succeeded became more open, critical, and reflective, and they incorporated EBN into daily practice in tailored and creative ways (e.g., EBN quizzes). Support from inspiring internal and external facilitators and driving forces within the team and facilitation by the managers, directors, and research networks were important for successfully developing an EBN culture in nursing teams. The EBN culture was further influenced by the team’s climate and composition. Continuous support and stimulation of EBN seem to be a prerequisite for developing and sustaining an EBN culture that eventually contributes to better quality of fundamental nursing care in nursing homes.

## Figures and Tables

**Table 1 ijerph-19-01733-t001:** Methods of data acquisition.

Measurement	Instrument	Subscales/Topics	Participants
** *Qualitative* **			
EBN culture and influencing factors	Individual interviews	EBN in daily practice Best possible care Barriers to EBN Facilitators for EBN Ideal picture of EBN in daily practice	Five from each organization: Internal facilitator Nursing team member Team manager Contact person from the academic nursing home network (director/manager) Medical care provider (elderly care physician/physician assistant/nurse practitioner)
Clinical leadership	Focus group interviews	Best possible care (vision and daily practice) Collaboration Continuous improvement Professional characteristics	Nurses (in training for) EQF 1–7
** *Quantitative* **			
Barriers to and facilitators for EBN	Barriers Scale (35 items) ^†^	Characteristics of the: Nurse Organization Innovation Communication	Nurses (in training for) EQF 6 and 7
EBN use, attitude, knowledge, and skills	Evidence-Based Practice Questionnaire EBPQ-ve (11 items) for EQF levels 1–4 EBPQ (25 items) ^‡^ for EQF levels 6 and 7	For EQF levels 1–4: Reflection Implementation Attitude Asking about client preferences For EQF levels 6 and 7: Practice Attitude Knowledge/skills	Nurses (in training for) EQF 1–4 Nurses (in training for) EQF 6 and 7
EBN attitude	Evidence-Based Practice Attitudes Scale EBPAS-ve (15 items) for EQF levels 1–4 EBPAS (15 items) ^§^ for EQF levels 6 and 7	Openness Divergence Appeal Requirements	Nurse (in training for) EQF 1–4 Nurses (in training for) EQF 6 and 7
Transformational leadership	Leadership Practice Inventory (LPI) (30 items)	Model the way Inspire a shared vision Challenge the process Enable others to act Encourage the heart	Internal facilitators

^†^ [12,23].^‡^ EBPQ-ve: [24]. EBPQ: [25]. ^§^ EBPAS-ve: [24]. EBPAS: [26,27,28,29].

**Table 2 ijerph-19-01733-t002:** Participating nursing team members in 12 participating teams.

	Pre-Intervention	Post-Intervention
EQF level 1	6	7
EQF level 2	34	34
EQF level 3	87	93
EQF level 4	47	48
EQF level 6	10	13
EQF level 7	2	2
In training for EQF level 1–4	34	71
In training for EQF level 6 and 7	12	11
Unknown	6	-
Total	238	279

**Table 3 ijerph-19-01733-t003:** EBPQ(-ve) and EBPAS(-ve).

Subscale ^†^	Nurses EQF 1–4		Nurses EQF 6 and 7	
	Mean (SD)		Mean (SD) ^‡^	
	Pre (N = 131)	Post (N = 119)	Pre (N = 16)	Post (N = 21)
**EBPQ-ve** (range 1–6)				
Attitude	4.4 (0.9)	4.1 (0.9) *		
Asking client preferences (1 item)	5.0 (0.8)	5.1 (0.8)		
Reflection	3.6 (0.7)	3.5 (0.8)		
Implementation	4.3 (0.9)	4.2 (0.9)		
*Total*	*4.2 (0.7)*	*4.1 (0.6)*		
**EBPQ** (range 1–7)				
Attitude			5.5 (1.0)	5.5 (0.9)
Asking client preferences (1 item)			5.2 (0.8)	5.7 (1.0)
Implementation			5.4 (0.8)	5.3 (1.1)
Knowledge and skills			5.4 (0.5)	5.2 (0.8)
*Total*			*5.4 (0.5)*	*5.3 (0.8)*
**EBPAS(-ve)** (range 1–4)				
Openness	2.5 (0.6)	2.5 (0.6)	2.9 (0.5)	2.7 (0.5)
Divergence	2.4 (0.5)	2.3 (0.6)	2.9 (0.5)	2.7 (0.5)
Appeal	3.2 (0.6)	3.1 (0.5)	3.0 (0.5)	3.1 (0.4)
Requirements	3.3 (0.7)	3.2 (0.6)	2.5 (0 8)	2.7 (0.7)
*Total*	*2.8 (0.4)*	*2.8 (0.4)*	*2.8 (0.3)*	*2.8 (0.3)*

* *p*-value < 0.05.^†^ Higher scores indicate better EBN use, attitude, knowledge, or skills. ^‡^ Differences not tested because of the small number of respondents.

**Table 4 ijerph-19-01733-t004:** Barriers Scale—Nurses at EQF levels 6 and 7.

Item	Subscale	Moderate or Great Barrier %	
		Pre (N = 16)	Post (N = 21)
There is insufficient time on the job to implement new ideas	Organization	94	62
Research reports/articles are not readily available	Communication	87	45
The facilities are inadequate for implementation	Organization	87	45
I feel isolated from knowledgeable colleagues with whom to discuss the research	Nurse	86	38
The nursing team is not supportive of implementation	Organization	80	65
Implications for practice are not made clear	Communication	79	60
The research is not relevant to nursing practice	Communication	79	32
I do not have time to read research	Organization	73	48
The workplace culture does not stimulate searching for and implementing research results	Organization	69	50
I am unaware of the research	Nurse	57	35
The literature reports conflicting results	Innovation	57	53
The conclusions drawn from the research are not justified	Innovation	54	33
The research is not reported clearly and readably	Communication	54	40
The amount of research information is overwhelming	None	54	60
I think the research I read has methodological inadequacies	Innovation	54	20
The statistical analyses are not understandable	Communication	50	50
I think the benefits of changing practice will be minimal	Nurse	50	30
I do not feel I have enough authority to change patient care procedures	Organization	44	35
The relevant literature is not compiled in one place	Communication	43	62
I feel results are not generalizable to own setting	Organization	36	33
I see little benefit for myself	Nurse	33	45
I am uncertain whether to believe the results of the research	Innovation	29	19
Management will not allow implementation	Organization	25	29
Physicians will not cooperate with implementation	Organization	20	16
I do not feel capable of evaluating the quality of the research	Nurse	20	29
There is not a documented need to change practice	Nurse	17	14
I do not see the value of research for practice	Nurse	13	28
I am not competent in searching the literature systematically	Nurse	13	15
I find it difficult to read English reports	Nurse	13	38
I am unwilling to change/try new ideas	Nurse	7	19

**Table 5 ijerph-19-01733-t005:** LPI ^†^ internal facilitators.

	Mean (SD) ^§^	
Subscale ^‡^	Pre-Intervention (N = 12)	Post-Intervention (N = 11)
Model the way	7.6 (0.5)	7.8 (0.6)
Inspire a shared vision	7.6 (0.7)	7.7 (0.8)
Challenge the process	7.3 (0.8)	8.0 (0.5)
Enable others to act	8.2 (0.4)	8.3 (0.6)
Encourage the heart	7.9 (0.7)	8.3 (0.6)
*Total*	*7.7 (0.5)*	*8.0 (0.4)*

^†^ Range 1–10. ^‡^ Higher scores indicate better transformational leadership qualities. ^§^ Differences not tested because of the small number of respondents.

## Data Availability

The datasets generated and/or analyzed during the current study are not publicly available due to the nature of the data and the relatively low number of participants but are available from the corresponding author on reasonable request.

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
