# Peer review of "Developing an Evidence-Based Nursing Culture in Nursing Homes: An Action Research Study"

_ijerph, 2022, doi:10.3390/ijerph19031733_

Round 1

Reviewer 1 Report

Thank you for preparing this manuscript which focuses on a priority area; improving the culture of evidence based practice within care homes. The work is robust, writing is clear, and the work demonstrates notable rigor and project management. My suggestions as an independent peer reviewer are below. They are minor and concern how you present the work more than anything.

The abstract does not do the work justice. I think it conflates evidence based nursing, culture change, your intervention and the procedures you undertook.  The results section of the abstract especially could provide more nuanced information.

My biggest concern is the lack of consultation with the literature on what is already known about organizational change, implementation, and evidence based culture change. Many of the facets you highlight as important in your results are already well established. A greater rationale and consultation with the literature prior to the research aim would be useful. I would recommend sharpening what this paper offers as an original contribution to the literature.

In the introduction I would suggest a more nuanced presentation of evidence based practice and organizational culture (a complex system). In the methods it is much clearer that you appreciate the system driver’s and the ‘art’ of evidence informed practice and the necessary interventions needed for change, but the introduction looks reductionist.

The methods could include greater clarity and an overview of methodological decisions and procedures e.g purposive sampling, qualitative analysis approach used (content, reflective thematic, framework etc). There are a range of aspects introduced in the practice development that are not introduced elsewhere. These seem important and require an introduction too e.g pre conditions.

Line 138 requires some clarification.

I would suggest a more concise and informative results section as the categories lack nuance e.g EBN culture, and the take home messages are hard to infer from the way it is presented. Some additional information on the analysis procedures would also be useful. Why does a sample size of 25 matter for pre post testing? Why those covariates and not others etc.

The final appraisal of strengths of the work also requires more nuance in my eyes. You talk about trustworthiness, unexpected findings, and triangulation as an afterthought. I would prefer a greater consideration of methodological rigor in the methods and for your findings to be compared to other work instead on offering your thoughts on why scales did not change as expected.  

Reviewer 2 Report

The issues raised by the authors are topical and important for nursing. The justification for taking the topic is sufficient. Conclusions correctly worded and the discussion is comprehensive and inclusive important items of current literature.

Reviewer 3 Report

This is a very interesting mixed methods action research study that measured change using evidence-based nursing practice in Danish nursing homes. The manuscript is well-written, and builds on their prior research in the area. The following comments are offered in order to strengthen the presentation.

Title: descriptive of the study

Abstract: spell out EBN the first time it is used in the manuscript. Under Results, the sentence "Most nursing teams created something of an EBN culture" is vague. You could say that "With varying degree, most nursing teams implemented elements (values, attitudes, and behaviors) of an EBN culture with appropriate leadership, advocacy, and training." This would give the reader a better idea of what happened.

Key words: external and internal facilitator and practice development are not MeSH terms. Consider eliminating these and using "change, organizational."

Introduction: well organized and cited.

Methods: Table 1 needs a more descriptive caption, perhaps "Methods of data acquisition." The researchers used Atlas.ti for qualitative data analysis, but very little of these data were presented. In addition, they didn't take advantage of the network analysis views to show the relationships between themes and codes. Incorporation of a "network view" would strengthen the qualitative section, and use of more in vivo quotes would assist the reader to better understand the participant's viewpoints.

At line 181, the authors identified 3 main categories, but highlighted only EBN culture. Please highlight and/or separate the other two categories for the reader's benefit.

Information in Table 4 will be very useful for future change efforts.

Discussion: Line 312 - 'Magnet' not 'magnetic.' Magnet is a program where nursing leaders successfully align their nursing strategic goals to improve the organization’s patient outcomes. See https://www.nursingworld.org/organizational-programs/magnet/ for more information.

The only limitation not explained is that there were no patient-related outcome measures used in the study, such as those described at https://www.cms.gov/Medicare/Quality-Initiatives-Patient-Assessment-Instruments/NursingHomeQualityInits/NHQIQualityMeasures, such as changes in influenza immunization, pressure ulcer, or antipsychotic medication use rates. An idea of the kind of problems that each nursing home needed to address might provide a glimpse into why a change culture was needed to be implemented.

References: generally in mdpi style.

Thank you for the opportunity to review this interesting study.
